# Live On the Hump: Self Knowledge Distillation via Virtual Teacher-Students Mutual Learning

## ABSTRACT

For solving the limitations of the current self knowledge distillation including never fully utilizing the knowledge of shallow exits and neglecting the impact of auxiliary exits' structure on the performance of network, a novel self knowledge distillation framework via virtual teacher-students mutual learning named LOTH is proposed in this paper. A knowledgeable virtual teacher is constructed from the rich feature maps of each exit to help the learning of each exit. Meanwhile, the logit knowledges of each exit are incorporated to guide the learning of the virtual teacher. They learn mutually through the well-designed loss in LOTH. Moreover, two kinds of auxiliary building blocks are designed to balance the efficiency and effectiveness of network. Extensive experiments with diverse backbones on CIFAR-100 and Tiny-ImageNet validate the effectiveness of LOTH, which realizes superior performance with less resource by the comparison with the state-of-the-art distillation methods. The code of LOTH is available on Github.

## CCS CONCEPTS

• **Do Not Use This Code → Generate the Correct Terms for Your Paper**; *Generate the Correct Terms for Your Paper*; Generate the Correct Terms for Your Paper; Generate the Correct Terms for Your Paper.

## KEYWORDS

Knowledge distillation, Self-distillation, Multi-Exits, Feature fusion

## 1 INTRODUCTION

In recent years, Convolutional Neural Networks (CNNs) have attracted considerable attention for their excellent performance in various computer vision tasks. However, remarkable performance of CNNs [23, 25] typically suffer from exorbitant computational resources and memory overheads, which makes it difficult to be applied in edge devices with limited resources. Knowledge Distillation (KD), as one of the most effective network compression techniques [5, 9, 18], aims to reduce the size of CNNs without changing its structure and ensure the feature representation capability of CNNs at the same time.

Traditional KD methods [6, 10, 12, 20, 21, 32] adopt two-stage offline training strategy, in which a high-capacity teacher network is pre-trained and then performs a one-way knowledge transfer

**Unpublished working draft. Not for distribution.**

*ACM MM, 2024, Melbourne, Australia*
© 2024 Copyright held by the owner/author(s). Publication rights licensed to ACM.
ACM ISBN 978-x-xxxx-xxxx-x/YY/MM
https://doi.org/10.1145/nnnnnnn.nnnnnnn

to low-capacity student network. Obviously, the performance of student network depends heavily on the fixed knowledge extracted by the static teacher network. However, sophisticated teacher networks are not always readily available. Specifically, networks with high accuracy are not necessarily capable of high-quality knowledge transfer [11]. Moreover, the two-phase training process is cumbersome, which leads to enormous computational costs and resource burdens. Owing to these limitations, end-to-end mutual learning strategies [3, 7, 14, 24, 27, 31] are in the spotlight. There is no explicit teacher-student role in this strategy, which allows multiple networks to be co-trained in a peer-teaching manner. Even though dynamic interactions between networks contribute to reducing semantic gaps, such aimless mutual transfer exists the risk of performance degradation when the capacity gaps of the two networks differ significantly. In addition, the cost of training multiple networks simultaneously remains prohibitive.

Self knowledge distillation (SKD) breaks the dilemmas, which extracts and learns knowledge from its own backbone. One kind of SKD shares the same shallow backbone to construct multiple peer auxiliary exits in the last stage of the backbone, which integrates logits of multi-exits as a teacher. However, multiple exits in the same location typically tend to cause homogenization [16], hurting the generalization ability of networks. Another kind of SKD adds auxiliary exits hierarchically in a deep supervised manner. It takes the deepest knowledge as supervised signal to guide the learning of shallow exits. Although it is remarkable both in the aspect of keeping the ability of backbones and reducing the training costs, it still suffers from two inherent limitations: 1) the complementary knowledge of shallow exits is never fully utilized, which decreases the effectiveness of distillation; 2) the design of auxiliary exits normally employs heuristic principles with a single and fixed architecture on all networks, which fails to strike a balance between effectiveness and efficiency.

To tackle these problems, a novel self knowledge distillation framework via virtual teacher-students mutual learning is proposed in this paper. Since the virtual teacher and the students are generated from the backbone itself, and they learn from each other, so we name the proposed self knowledge distillation as LOTH (Live on the hump). Each exit in a network produces a response as a logit, which is though as a student. The feature maps from all the exits are used to build a virtual teacher, which concentrates all the important knowledge of all the exits in a network. Specifically, the feature maps of all exits perform cross-channel information interaction in both global and local manners. It captures finer-grained identifiable information to adaptively enhance the fusion representation, giving us a knowledgeable virtual teacher. In order to learn more meaningful knowledge, a mutual learning between the virtual teacher and multi-exits of the backbone is designed according to the classification loss and distillation loss. In addition, for balancing the efficiency and effectiveness, two kinds of bottleneck building

blocks are designed to capture the hidden information of the backbone. Extensive experiments with diverse backbones on CIFAR-100 and Tiny-ImageNet datasets show that LOTH improves the top-1 accuracy of classification by 3.81% to 6.58% with less resources compared with diverse backbones, and the top-1 accuracy achieved by LOTH is 3.81% to 6.58% higher than the state-of-the-art distillation methods with equivalent amount of parameters. In summary, the main contributions of this paper are as follows:

- A novel self knowledge distillation framework named LOTH is proposed, which views each exit as a student and fuses knowledge from auxiliary exits to build a knowledgeable virtual teacher. The bidirectional mutual learning between the virtual teacher and students contributes significantly to the capabilities of multi-exits learning and significant gains with few training overhead.
- An efficient adaptive feature fusion is designed to adequately extracts important knowledge from each exit, which enhances the feature maps by channel and spatial attention, resulting in a powerful and knowledgeable virtual teacher.
- The impact of different scales of auxiliary exits' structures are explored on the model's performance, customizing two well-designed auxiliary blocks to balance the efficiency and effectiveness based on the size of the network.

## 2 RELATED WORKS

### 2.1 Knowledge Distillation

In the knowledge distillation framework, there are always other networks to provide addition supervised signals. The vanilla KD pioneered by [6] utilizes the fixed soft logit knowledge of teacher network to guide the learning of the student network. To promote the efficiency of logit knowledge, DKD [32] thoroughly explores the logit distillation and decouples it into target class knowledge distillation and non-target class knowledge distillation. Moreover, PESF-KD [21] further reduces time-consuming pre-trained teacher network and enables adaptive knowledge transfer by tuning the soft logit of the teacher network. In addition to these explorations of logit knowledge described above, there are also efforts to captures intermediate information of teacher networks. AT [12] narrows the distance gap of attention maps between teacher and student networks. RKD [20] facilitates knowledge transfer of structural relationships between sample instances through distance-wise and angle-wise losses. Besides, DIST [10] captures the intrinsic interclass relations of teacher to enrich correlation-based knowledge. However, all the above approaches adopt a two-stage training strategy, where high-quality teacher networks are pre-trained to provide priori knowledge for the low-capacity student network to emulate, which is tedious and complicated.

To enhance the efficiency of KD, numerous works start to focus on online mutual learning of multiple networks, which simplifies the training process to one-stage. DML [31] encourages each peer network to learn from each other through logit distillation. DCCL [24] additionally introduces intermediate cross-layer feature supervision to facilitate dynamic interactions between networks. MCL [27] enables multiple networks to conduct mutual contrastive learning to achieve interaction and transfer of contrastive distributions. Aside from these inter-network transfer methods, there exist ensemble strategies to generate an intermediary for collaborative learning, which in turn guides the learning of each network. KDCL [3] ensemble soft logits from multiple networks for high-quality supervision transferred to each peer networks. While DualNet [7] sums the feature maps of all peer networks to obtain a robust teacher classifier. Moreover, EML [14] provides a lightweight adaptive feature fusion module that concatenates features from all sub-networks.

### 2.2 Self Knowledge Distillation

Instead of relying on extra networks, self Knowledge distillation extracts knowledge from its own backbone in the form of multiple auxiliary exits. ONE [33] adopts simple gating mechanisms to dynamically integrate the logits of multiple peer exits, where each exit is treated as a student and the fused logit acts as a teacher. OKD-Dip [1] adopts self-attention mechanisms to implement two-level distillation with multiple peers exits and a group leader. Moreover, Zhu et al. [34] further extend OKDDip, which utilizes a concise feature learning to provide precise weights for peer exits. AHBF-OKD [2] constructs diverse peers to promote knowledge diversity and employs hierarchical fusion to learn complementary knowledge. All of these approaches described above share the same shallow backbone to construct multiple peer exits, which suffers from a fatal homogeneity defect.

To further improve the efficiency of self Knowledge distillation, substantial works pay more attention to the strategy of hierarchical shared shallow backbone. BYOT [29] adds multiple auxiliary exits hierarchically to capture the hidden knowledge. ECSD [28] proposes additional attention modules at the connection between the backbone and auxiliary exits to capture richer features. However, these approaches all view the deepest layer of the backbone as teacher to supervise the learning of shallow exits, which neglects the exploitation of complementary knowledge provided by shallow exits. To this end, BEED [13] proposes an artificially regulated ensemble strategy to merge the knowledge of all exits to instruct each exits, which is mechanical and inconvenient for different network architectures and datasets. KFD [15] adopts an adaptive fusion strategy to integrate feature maps from earlier backbones, which possesses relative weaker semantic information than exit. Moreover, DTSKD [17] additionally considers the past learning history to further enrich the form of the supervised signal. In contrast to these approaches, our proposed LOTH adequately integrates complementary feature and logit knowledge from early exits to facilitate mutual learning across multi-exits, which possesses excellent generalization and robustness.

## 3 METHOD

### 3.1 Problem Formulation

Suppose a convolutional neural network has $k$ stages. Each stage is composed of multiple specific building blocks that can be thought as feature extractor, denoted as $\mathcal{G}_1, \mathcal{G}_2, ... \mathcal{G}_k$. Given $N$ training samples $\mathcal{X} = \{x_n\}_{n=1}^{N}$, the corresponding ground-truth labels with $M$ classes are denoted as $\mathcal{Y} = \{y_n \in \mathbb{R}^M\}_{n=1}^{N}$. The purpose is to train such $k$-stage network by using these $N$ samples to get a same structure network with more robust classification ability and fewer

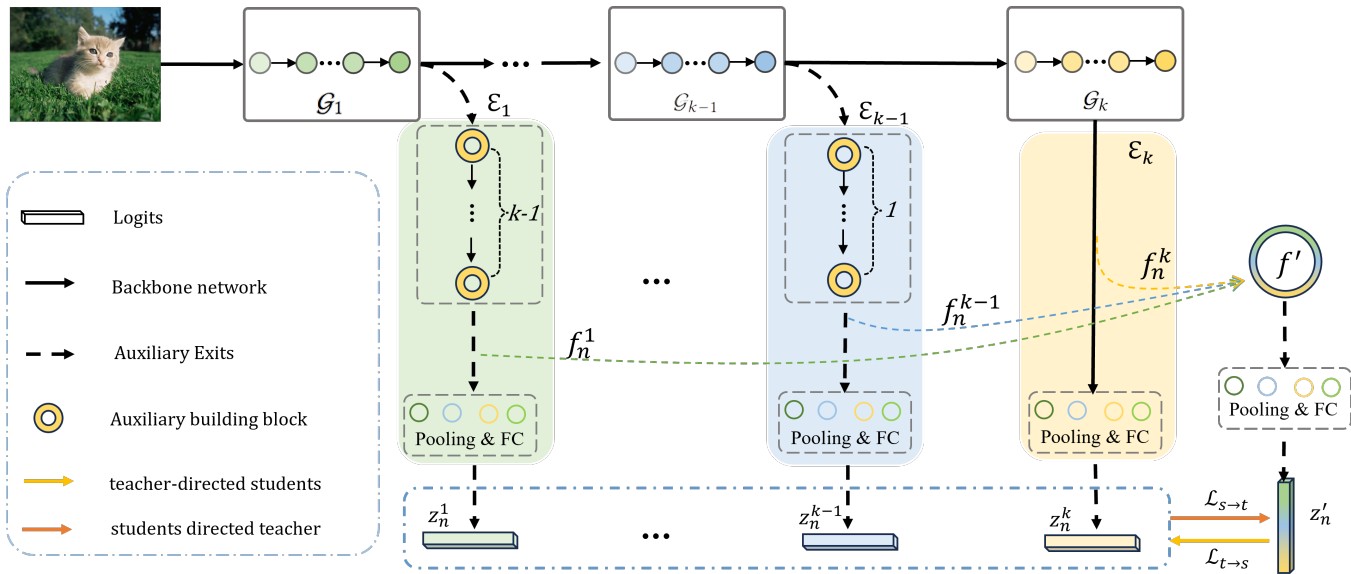

Figure 1: The overall architecture of LOTH. The solid lines represent the backbone network, while the dashed lines indicate auxiliary exits, which can be removed during the inference phase.

parameters. Fig. 1 shows the whole framework of LOTH. Take the sample $x_n$ as an example, it is input into the backbone network, $k$ logits are generated from the $k-1$ auxiliary exits and one main backbone exit. At the same time, a virtual teacher is built based on these $k$ exits, bringing in $z'_n$. Then $z'_n$ and these $k$ logits learn the knowledge of backbone mutually according to the loss $\mathcal{L}_{s \to t}$ and the loss $\mathcal{L}_{t \to s}$.

## 3.2 Response of Auxiliary Exits

For the network with $k$ stages, $k-1$ auxiliary exits are added at early stages. Let's use $\mathcal{E}_1$ to denote the auxiliary exit in the first stage, similarly, there are $\mathcal{E}_2...\mathcal{E}_k$ for other stages, where $\mathcal{E}_k$ is the default exit of backbone. For the sample $x_n$, it will pass through feature extractors in the backbone network and the corresponding exits to generate logits $z_n^1, z_n^2, ..., z_n^k$. Such process can be simply formed as:

$$
\begin{aligned}
z_n^1 &= \mathcal{E}_1 \circ \mathcal{G}_1(x_n), \\
z_n^2 &= \mathcal{E}_2 \circ \mathcal{G}_2 \circ \mathcal{G}_1(x_n), \\
&\cdots, \\
z_n^k &= \mathcal{E}_k \circ \mathcal{G}_k \circ \mathcal{G}_{k-1} \circ \cdots \circ \mathcal{G}_1(x_n).
\end{aligned}
\tag{1}
$$

The auxiliary exits $\{\mathcal{E}^i\}_{i=1}^{k-1}$ play the role of providing additional training objectives in LOTH. Each auxiliary exit comprises two parts: a feature transformation and a classifier. The classifier provides the foundation for probabilistic prediction, which is implemented by a fixed pooling layer and a fully connected (FC) layer. The feature transformation composes of multiple building blocks with downsampling, which ensures that the sizes of output features from the shallow auxiliary exits are identical to the $k$-th stage of the backbone. Obviously, the feature transformation dominates the

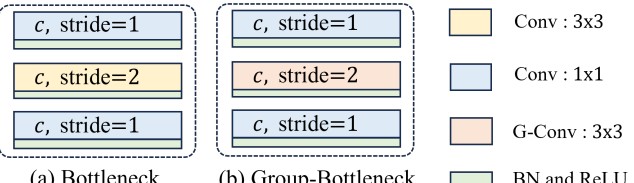

(a) Bottleneck  (b) Group-Bottleneck

Conv : 3x3
Conv : 1x1
G-Conv : 3x3
BN and ReLU

Figure 2: The structure of a single auxiliary building block, where $c$ denotes the number of output channels, and G-Conv means group convolution (depth-wise convolution).

auxiliary exits. Therefore, the design of auxiliary building blocks is extremely crucial.

Previous researches [13, 29] have demonstrated that Bottleneck block originated from ResNet [4] is an effective architecture to improve the performance of exits. However, the single employment of a fixed architecture ignores the impact of the relative size of the auxiliary exits on the performance of the backbone network, which is not optimal across all networks.

Hence, two kinds of auxiliary building blocks are designed at here. One is Bottleneck, and the other is Group-Bottleneck. They are tailored according to the scale of the backbone network. As illustrated in Fig 2(a), the channel expansion and residual connection are removed from the standard Bottleneck [4], which retains the powerful extraction capabilities and reduces computational burden. However, Bottleneck drastically enlarges the scale of auxiliary exits in the lightweight networks, slowing the training speed. Group-Bottleneck designed in Fig. 2(b), uses a group convolution (depth-wise convolution) rather than a regular convolution in the middle layer of the Bottleneck block, significantly reducing the scale of auxiliary exits. One thing should be pointed out is that the

Anonymous Authors

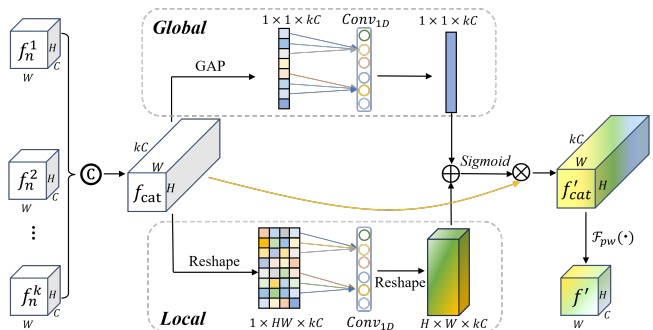

**Figure 3: An illustration of virtual teacher generation. The features of $k$ exits are fused to be $f^{'}$, which will be mapped to be $z_n^{'}$**

number of auxiliary building blocks in each auxiliary exit depends on the stage of the backbone network. Taking the $k = 3$ as an example, the number of building blocks in $\mathcal{G}_1$ is 2, the number of building blocks in $\mathcal{G}_2$ is 1, and the number of building blocks in $\mathcal{G}_3$ is 0, i.e., the default backbone exit.

### 3.3 Generation of Virtual Teacher

The auxiliary building blocks in each auxiliary exit always generate rich semantic features, which should be employed to help exits learn the knowledge of backbone network.

Let $f_n^i \in \mathbb{R}^{H \times W \times C}$ $(i = 1, 2, \dots k)$ denote the features generated from $\mathcal{E}_i$, where $C$ means the number of channels and $H \times W$ tells us the size of feature maps. By simply concatenating $f_n^i$ along the channel dimension, $f_{cat} \in \mathbb{R}^{H \times W \times C^{'}}$ is obtained, where $C^{'} = kC$. Subsequently, $f_{cat}$ is transformed along two parallel paths simultaneously. On one path, $f_{cat}$ is globally transformed based on channels, giving the global feature context $f_g$ that tends to emphasize large objects with coarse granularity. $f_g$ is calculated by:

$$f_g = \mathbb{C}_{1D}(\text{GAP}(f_{cat})), \quad f_g \in \mathbb{R}^{1 \times 1 \times C^{'}}, \tag{2}$$

where $\mathbb{C}_{1D}$ indicates 1D-Convolution, and GAP stands for global average pooling that captures the global distribution of channel dimensions.

On the other path, $f_{cat}$ is locally transformed, giving the local feature context $f_l$ that focuses on subtle details of small object. $f_l$ is calculated by:

$$f_l = \mathcal{R}(\mathbb{C}_{1D}(\mathcal{R}(f_{cat}))), \quad f_l \in \mathbb{R}^{H \times W \times C^{'}}, \tag{3}$$

where $\mathcal{R}$ denotes the reshape transformation. There are a total of two transformations, where the first reshape transforms the input feature map of size $H \times W$ into a scalar for 1D-convolution in the channel direction, and the second reshape carries out a restoration. Note that the local feature context has the same size with the input, which can be viewed as a fine-scale perception of spatial location information. Moreover, the employment of 1D-Convolution facilitates information interaction across channels and ensures low complexity.

As for the kernel size $\mathcal{K}$ of 1D convolution, we adopt an adaptive strategy in [26] to determine:

$$\mathcal{K} = \varphi(C^{'}) = \left| \frac{log_2(C^{'})}{\gamma} + \frac{b}{\gamma} \right|_{odd} \tag{4}$$

where $|Q|_{odd}$ represents the nearest odd number of $Q$, $\gamma$ and $b$ are constant. In this work, we set $\gamma = 2$ and $b = 1$ for all experiments.

After obtaining the global feature context and local feature context, $\bar{f}_{cat}$ is enhanced by:

$$f_{cat}^{'} = f_{cat} \otimes \psi(f_g \oplus f_l), \quad f_{cat}^{'} \in \mathbb{R}^{H \times W \times C^{'}} \tag{5}$$

where $\psi(\cdot)$ indicates Sigmoid function to generate the final weight map, $\oplus$ and $\otimes$ denote the broadcasting addition and element-wise multiplication, respectively. Moreover, a lightweight point-wise convolution $\mathcal{F}_{pw}(\cdot)$ is utilized to further facilitate feature fusion of multi-exits and reduce the channel dimension. Eventually, the fused representation $f^{'}$ with strong discrimination and generalization capabilities is obtained, and then fed into classifier for the overall classification, which plays the role of a virtual teacher to help exits learn knowledge of backbone network.

### 3.4 Virtual Teacher-Students Mutual Learning

Now, from $k$ exits, $k$ logits have been generated. And a virtual teacher $z_n^{'}$ has been created from the advanced feature maps of all auxiliary exits, which can be treated as additional supervised signals to instruct the learning of each exit. The training loss $\mathcal{L}_{s \rightarrow t}$ of auxiliary exits is expressed as:

$$L_{s \rightarrow t} = \sum_{i=1}^{k} \mathcal{L}_c(y_n, z_n^i) + \mathcal{L}_d(z_n^{'}, z_n^i), \tag{6}$$

where $\mathcal{L}_c$ denotes the fundamental classification loss, which is defined as in terms of the Cross-Entropy (CE) between the probability distribution and the ground-truth label of the samples:

$$\mathcal{L}_c(y_n, z_n^i) = CE(y_n, \sigma(z_n^i)), \tag{7}$$

where $\sigma(\cdot)$ is the softmax function. The distillation loss $\mathcal{L}_d$ employs the Kullback-Leibler (KL) divergence to measure the soft distribution difference between the target and sample:

$$\mathcal{L}_d(z_n^{'}, z_n^i) = \tau^2 \cdot KL(\sigma(z_n^{'}/\tau), \sigma(z_n^i/\tau)) \tag{8}$$

where $\tau$ is the temperature coefficient. As $\tau$ increases, the probability distribution gets smoother.

Considering that early exits of the network typically have relatively weak extraction capabilities, it is difficult to capture sufficient information directly from low entropy ground-truth labels. Therefore, inspired by [13], we further control the proportion of classification loss $\mathcal{L}_c$ and distillation loss $\mathcal{L}_d$ in Eq 6 to enable early exits to learn more subtle knowledge from the soften distillation distribution, which can be reformulated as:

$$L_{s \rightarrow t} = \sum_{i=1}^{k} (2 - \alpha^{k+1-i}) \cdot \mathcal{L}_c(y_n, z_n^i) + \alpha^{k+1-i} \cdot \mathcal{L}_d(z_n^{'}, z_n^i), \tag{9}$$

where the balance factor $\alpha$ satisfies $\alpha > 1$ and $\alpha^{k+1-i} < 2$, which guarantees the early exits to be always assigned to more distillation weights.

**Table 1: Top-1 classification accuracy and parameter statistics of LOTH on CIFAR-100.**

| Networks | Baseline | Exit1 | | Exit2 | | Exit3 | | Exit4 | | Fusion |
|---|---|---|---|---|---|---|---|---|---|---|
| | | Acc(%) | Param(M) | Acc(%) | Param(M) | Acc(%) | Param(M) | Acc(%) | Param(M) | |
| VGG16 | 73.54 | 76.31 | 6.64 | 76.46 | 7.43 | 76.87 | 10.58 | **76.94** (↑ 3.40) | 15.30 | 78.68 |
| VGG19 | 73.34 | 75.23 | 6.64 | 75.73 | 8.02 | **76.09** | 13.53 | 76.06 (↑ 2.72) | 20.61 | 77.83 |
| ResNet18 | 77.65 | 78.47 | 3.82 | 79.53 | 4.17 | 80.90 | 5.58 | **81.24** (↑ 3.59) | 11.22 | 82.08 |
| ResNet34 | 78.00 | 78.37 | 3.89 | 80.53 | 4.84 | 81.84 | 10.97 | **81.96** (↑ 3.96) | 21.33 | 82.77 |
| MobileNetV1 | 73.40 | 76.99 | 2.22 | 77.40 | 2.23 | 78.62 | 3.30 | **79.02** (↑ 5.62) | 3.31 | 80.62 |
| MobileNetV2 | 72.22 | 76.46 | 2.93 | 76.79 | 2.78 | 76.95 | 2.45 | **77.51** (↑ 5.29) | 2.35 | 80.13 |
| ShuffleNetV1 | 71.39 | 74.31 | 1.91 | 75.31 | 1.89 | **76.69** | 1.85 | 76.35 (↑ 4.96) | 1.01 | 79.06 |
| ShuffleNetV2 | 71.85 | 73.04 | 1.51 | 74.82 | 1.52 | 75.93 | 1.68 | **76.03** (↑ 4.18) | 1.36 | 79.14 |

To make full use of logit knowledges, we further incorporate the logits at each exit to construct a powerful ensemble logit $\bar{z}_n$:

$$\bar{z}_n = \frac{1}{k}\sum_{i=1}^{k} z_n^i, \qquad (10)$$

which is utilized to guide the learning of the virtual teacher:

$$\mathcal{L}_{t\to s} = \mathcal{L}_c(y_n, z_n') + \mathcal{L}_d(\bar{z}_n, z_n'). \qquad (11)$$

Bidirectional knowledge communication between the virtual teacher and auxiliary exits facilitates the learning across exits. In the end, the total training objective can be obtained by summing the training loss of $\mathcal{L}_{s\to t}$ and $\mathcal{L}_{t\to s}$:

$$\mathcal{L}_{total} = \mathcal{L}_{s\to t} + \mathcal{L}_{t\to s}. \qquad (12)$$

## 4 EXPERIMENTS

### 4.1 Experiments Setup

*4.1.1 Datasets.* For evaluating the proposed method, two multi-categorical benchmark datasets are used: (1) *CIFAR-100*, a natural image dataset, comprising 50K / 10K training / test samples drawn from 100 classes, sized at $32 \times 32$ pixels per sample; (2) *Tiny-ImageNet*, a more challenging dataset, containing 200 classes, each class has 5K / 500 training / test samples with $64 \times 64$ pixels. In the actual evaluation process, the images in Tiny-ImageNet are resized to be the same as CIFAR-100. Moreover, data enhancement with horizontal flipping and random cropping are performed on the training samples of both datasets.

*4.1.2 Backbones.* To validate the effectiveness of the proposed LOTH, multiple backbone networks are adopted. They are popular architectures ResNet [4] and VGG [23], and lightweight networks MobileNetV1 [8], MobileNetV2 [22], ShuffleNetV1 [30], and ShuffleNetV2 [19]. Given the low image resolution of both datasets, these backbone network architectures are slightly modified by dropping the first pooling layer and shrinking the stride and kernel size of the convolutional layer.

*4.1.3 Implementation Details.* All the networks are implemented by Pytorch on GeForce RTX2080Ti GPU with 11 GB memory. The stochastic gradient descent (SGD) optimizer with momentum of 0.9 and weight decay of 5e-4 are utilized to optimize networks for 200 epochs. We set the initial learning rate to 0.1 for both datasets, which is divided by 10 at 75th, 130th and 180th epochs. We set the mini-batch size to 128, the temperature coefficient $\tau$ to 3, and the balancing factor $\alpha$ to 1.15 by default during all training procedures.

### 4.2 Comparing with Standard Training

The experimental results of LOTH on CIFAR-100 and Tiny-ImageNet in comparison with various backbones are presented in Table 1 and Table 2, respectively, where the adapted Bottleneck (Bn) block is utilized for both regular-sized ResNet and VGG architectures, and the Group-Bottleneck (G-Bn) block is adopted for four lightweight networks. The last columns 'Fusion' shows the top-1 accuracy obtained by the virtual teacher in LOTH. These two tables show that: (1) Compared to the standard training, the whole performance of all backbone networks benefits tremendously from LOTH. Specifically, with the same capacity, our approach surpasses the baselines by margins of 2.72% to 5.62% on CIFAR-100 and profound improvements of 3.81% to 6.58% on Tiny-ImageNet. (2) LOTH exceeds the performance of almost all baselines at the shallowest exit. Importantly, the shallowest exit has extremely low parameter advantages in some regular sized networks. Taking VGG19 at exit1 as example, LOTH achieves 2.39% and 6.91% gains on CIFAR-100 and Tiny-ImageNet with barely 1/3 of the backbone parameters, respectively. (3) The performance of each exit is not always increase with depth, and the optimal results are concentrated in the third and fourth exits. However, this is not a defect, but rather an advantage for self knowledge distillation. Owing to the hierarchical shared backbone, we can adaptively choose classifiers of different depths to trade-off dynamic accuracy and efficiency demands in real-world deployments. (4) On all backbone networks, the performance of the virtual teacher far exceeds that of other exits, indicating that the virtual teacher extracts richer and more meaningful semantic information. Therefore, it is reasonable to be taken as a virtual teacher. All these commonalities across different datasets and backbones confirm the strong robustness and generalization of LOTH.

### 4.3 Comparing with State-of-art Online Distillations

To validate the effectiveness of LOTH, a series of advanced online distillation methods are compared, which is presented in Table 3. The online distillation methods are further categorized into three strategies: (1) Multi-networks mutual learning: DML [31], EML[14],

**Table 2: Top-1 classification accuracy and parameter statistics of LOTH on Tiny-ImageNet.**

| Networks | Baseline | Exit1 | | Exit2 | | Exit3 | | Exit4 | | Fusion |
|---|---|---|---|---|---|---|---|---|---|---|
| | | Acc(%) | Param(M) | Acc(%) | Param(M) | Acc(%) | Param(M) | Acc(%) | Param(M) | |
| VGG16 | 50.38 | 54.22 | 6.70 | 54.74 | 7.48 | **55.02** | 10.63 | 54.63 (↑ 4.25) | 15.35 | 57.97 |
| VGG19 | 48.34 | 54.45 | 6.70 | 54.88 | 8.07 | **54.89** | 13.58 | 54.33 (↑ 5.99) | 20.66 | 57.76 |
| ResNet18 | 57.20 | 59.34 | 3.87 | 61.25 | 4.22 | 61.49 | 5.63 | **61.65** (↑ 4.45) | 11.27 | 63.92 |
| ResNet34 | 59.54 | 59.10 | 3.94 | 61.13 | 4.89 | 62.39 | 11.02 | **63.35** (↑ 3.81) | 21.38 | 64.96 |
| MobileNetV1 | 52.64 | 56.44 | 2.33 | 57.00 | 2.33 | 58.48 | 3.41 | **59.22** (↑ 6.58) | 3.41 | 61.81 |
| MobileNetV2 | 51.61 | 55.27 | 3.06 | 55.92 | 2.91 | 56.66 | 2.58 | **57.04** (↑ 5.43) | 2.48 | 59.98 |
| ShuffleNetV1 | 51.25 | 53.80 | 2.01 | 55.56 | 1.98 | **56.15** | 1.94 | 55.68 (↑ 4.43) | 1.11 | 58.47 |
| ShuffleNetV2 | 51.84 | 49.91 | 1.61 | 52.45 | 1.62 | 55.61 | 1.78 | **56.47** (↑ 4.63) | 1.46 | 58.11 |

**Table 3: Top-1 accuracy comparison of LOTH with state-of-art online distillation methods on CIFAR-100.**

| Methods | Years | ResNet18 | | VGG16 | |
|---|---|---|---|---|---|
| | | Acc(%) | Gain(↑) | Acc(%) | Gain(↑) |
| DML | 2018 | 78.97 | 1.32 | 74.40 | 0.86 |
| EML | 2023 | 79.75 | 2.10 | 74.60 | 1.06 |
| DCCL | 2023 | - | - | 76.10 | 2.56 |
| ONE | 2018 | 78.89 | 1.24 | 74.38 | 0.84 |
| OKDDip | 2020 | 79.83 | 2.18 | 74.85 | 1.31 |
| AHBF | 2023 | 78.82 | 1.17 | 75.08 | 1.54 |
| BYOT | 2019 | 78.59 | 0.94 | 74.98 | 1.44 |
| ECSD | 2021 | 79.61 | 1.96 | 75.56 | 2.02 |
| BEED | 2022 | 80.68 | 3.03 | 75.09 | 1.55 |
| KFD | 2023 | 79.48 | 1.83 | 75.04 | 1.50 |
| DTSKD | 2024 | 80.46 | 2.81 | 76.72 | 3.18 |
| LOTH | 2024 | 81.24 | 3.59 | 76.94 | 3.40 |

and DCCL [24]; (2) SKD with peer exits: ONE [33], OKDDip [1] and AHBF-OKD[2]; (3) SKD with hierarchical exits: BYOT [29], ECSD [28], BEED [13], KFD [15], and DTSKD [17]. To ensure fairness and consistency, we take the average of the two sub-networks as the result of multi-networks mutual learning, while for SKD, we take the accuracy of the main backbone network. It is observed that our LOTH achieves the optimal accuracy than all other online approaches in both backbones. Specifically, LOTH realizes gains of 0.58% to 2.56%, 0.22% to 2.58% on ResNet18 and VGG16, respectively. Even compared with SKD with equivalent architectures, LOTH still achieves an average of 1.48% and 1.46% improvement on ResNet18 and VGG16, respectively.

In addition, we additionally select several SKD methods with equivalent architecture to LOTH for further comparing the top-1 accuracy and the amount of parameters on Tiny-ImageNet. From Table 4, we can observe that (1) In the case of a large gaps of parameters, the ECSD with average ensemble strategy of multi-exits logits outperforms the BYOT which takes the deepest exit as the supervised signal. This suggests that using only the deepest knowledge as a guide limits the learning ability of the model. (2)

LOTH at the Exit1 with 1/3 of backbone parameters outperforms BYOT and ECSD at the Exit4, which demonstrates the importance of auxiliary building block structure. (3) Compared with BEED having the same parameters, LOTH still realizes a gain of 0.22% to 1.00% in multiple exits, which indicates the effectiveness of the virtual teacher-students mutual learning in LOTH.

## 4.4 Evaluation of Auxiliary Building Blocks

Figure 4 shows the performance of LOTH under different auxiliary building blocks on: regular-size networks (ResNet, VGG16) and lightweight networks (MobileNetV1, ShuffleNetV1), where Bn, G-Bn, and DSC stand for Bottleneck, Group-Bottleneck and Depth Separable Convolution utilized in ECSD [28], respectively. It can be observed: (1) With the similar lightweight design, our G-Bn block adds few training overheads yet achieves a significant benefit on all exits than DSC. (2) It is reasonable to treat the feature fusion as a teacher to instruct each exit. This is mainly reflected in the fact that the performance of fusion far exceeds the other exits under all three construction blocks, especially on DSC, whose performance varies greatly across exits. (3) The use of Bn with standard convolution achieves optimal main backbone performance at both regular-size networks, but is equal to or even lower than G-Bn with lightweight convolution on the two lightweight networks. More importantly, using BN to construct auxiliary exits on shuffleNetV1 imposes 34× training overhead, which slows down training significantly. Therefore, it is necessary to design two well-designed auxiliary blocks to balance the effectiveness and efficiency according to the scale of backbones.

## 4.5 Evaluation of Feature Fusion Mechanisms

To validate the effectiveness of the feature fusion in the step of virtual teacher generation, we further compare it on Tiny-ImageNet with three prevailing feature fusion strategies: Summation [7], Concatenation [14], and Sample-based Attention mechanisms (shorted as Att-Sample) [15]. From Table 5, we can note that the simple summation gets the worst performance, with a significant gap of 1.97% to the penultimate performance, which indirectly indicates the existence of tremendous semantic gaps between multi-exits. Moreover, our fusion results exceeds the sub-optimal performance by 0.51 among numerous advanced fusion strategies, which indicates our fusion possesses a stronger feature extraction capability to

**Table 4: Top-1 accuracy and parameter statistics of LOTH VS. advanced SKDs with hierarchical exits in ResNet18 on Tiny-ImageNet.**

| Methods | Supervision | Exit1 | | Exit2 | | Exit3 | | Exit4 | | Fusion |
|---------|-------------|-------|-----|-------|-----|-------|-----|-------|-----|--------|
| | | Acc(%) | Params(M) | Acc(%) | Params(M) | Acc(%) | Params(M) | Acc(%) | Params(M) | |
| BYOT | Deepest | 44.83 | 2.91 | 53.26 | 3.47 | 57.80 | 5.63 | 58.97 | 11.27 | 61.32 |
| ECSD | Ensemble-Avg | 47.09 | 0.44 | 53.91 | 0.97 | 57.34 | 3.08 | 59.03 | 11.27 | 60.51 |
| BEED | Ensemble-Weight | 59.20 | 3.87 | 60.25 | 4.22 | 60.52 | 5.63 | 61.11 | 11.27 | 64.13 |
| LOTH | Mutual Learning | **59.44** | 3.87 | **61.25** | 4.22 | **61.49** | 5.63 | **61.65** | 11.27 | 63.92 |

**Table 5: Top-1 accuracy comparison of different fusion mechanism in MobileNetV1 on Tiny-ImageNet.**

| Fuse type | Exit1 | Exit2 | Exit3 | Exit4 | Fusion |
|-----------|-------|-------|-------|-------|--------|
| Summation | 54.65 | 55.33 | 56.70 | 56.78 | 59.45 |
| Concatenation | 56.79 | 57.49 | 58.40 | 58.75 | 60.93 |
| Att-Sample | 56.87 | 57.19 | 58.79 | 58.84 | 61.30 |
| Ours | 56.44 | 57.00 | 58.48 | **59.22** | **61.81** |

mitigate the feature semantic gap between multi-exits. Importantly, likewise for the attention mechanism, our strategy still outperforms Att-Sample strategy by 0.38% on the main backbone exit.

**Table 6: Ablation study of global and local transformation on CIFAR-100.**

| Method | ResNet18 | | MobileNetV1 | | ShuffleNetV1 | |
|--------|----------|----------|-------------|----------|--------------|----------|
| | Acc(%) | Gain(↑) | Acc(%) | Gain(↑) | Acc(%) | Gain(↑) |
| w/o G & L | 80.78 | 3.13 | 78.66 | 3.37 | 75.43 | 4.04 |
| w G | 81.01 | 3.36 | 78.84 | 3.55 | 75.92 | 4.53 |
| w L | 80.53 | 2.88 | 78.33 | 3.04 | 75.88 | 4.49 |
| w G & L | 81.24 | 3.59 | 79.02 | 3.73 | 76.35 | 4.96 |

## 4.6 Ablation study

*4.6.1 Ablation of Global and Local feature context.* In the virtual teacher generation, global (G) and local (L) feature contexts work together to mitigate the feature semantic gap between multiple exits. To further explore the complementary of each component, careful deconstruction is performed and examined on multiple backbone architectures. From Table 6, we can observe: (1) Compared to the baseline of the first row, our fusion mechanism achieve significant performance gains with a range of 0.36% to 0.92% across the three backbones, which further indicates that our fusion can capture more discriminative information to mitigate semantic gaps. (2) Compared to local feature context, global feature context with a wide field provides a greater contribution to information capture, outperforming the case of using only local feature context by an average of 0.33%. (3) The information in the global and local feature contexts is complementary, pairing the two gives better performance. Although the performance of local feature context alone is lower than the baseline in ResNet18 and MobileNetV1, the

combination of global feature context exceeds the optimal value by 0.18% to 0.43%.

**Table 7: Ablation study of virtual teacher-students mutual learning on CIFAR-100.**

| Backbone | Method | Exit1 | Exit2 | Exit3 | Exit4 | Fusion |
|----------|--------|-------|-------|-------|-------|--------|
| VGG16 | w/o $\mathcal{L}_d$ | 72.47 | 74.88 | 75.75 | 75.77 | 77.81 |
| | w/o $\mathcal{L}_{t \to s}$ | 75.70 | 75.17 | 76.13 | 76.15 | – |
| 73.54 | w/o $\alpha$ | 75.54 | 75.88 | 76.69 | 76.63 | 78.38 |
| | LOTH | 76.31 | 76.46 | 76.87 | **76.94** | 78.68 |
| MobileNetV1 | w/o $\mathcal{L}_d$ | 71.77 | 73.25 | 74.92 | 74.96 | 78.82 |
| | w/o $\mathcal{L}_{t \to s}$ | 76.62 | 77.39 | 77.91 | 78.29 | – |
| 73.40 | w/o $\alpha$ | 76.00 | 76.78 | 78.35 | 78.51 | 80.33 |
| | LOTH | 76.99 | 77.40 | 78.62 | **79.02** | 80.62 |

*4.6.2 Ablation of virtual teacher-students mutual Learning.* The virtual teacher-students mutual learning plays the role of facilitating information interaction between fusion and multi-exits in LOTH. To explore the effectiveness of each component of this learning, as shown in Table 7, we perform ablation comparisons with the following three cases: (1) w/o $\mathcal{L}_d$ : Only the fundamental classification loss $\mathcal{L}_c$ in $\mathcal{L}_{s \to t}$ and $\mathcal{L}_{t \to s}$ is retained. Observing the first line of VGG16 and MobileNetV1, we can discover that even without distillation loss involved, our approach still achieves a remarkable gain of 2.23 and 1.56 compared to the baselines, respectively, which demonstrates the effectiveness of the hierarchical multi-exits architecture. (2) w/o $\mathcal{L}_{t \to s}$: Only logit knowledge of each exit is utilized to obtain the ensemble logit $\tilde{z}_n$, which served as supervised signals to guide multi-exits learning. It is not difficult to observe that without the involvement of feature map knowledge, there is a significant performance degradation in two backbones, which further validates the importance of virtual teacher. (3) w/o $\alpha$: Both the classification loss and distillation loss at each exit are signed the same weight of 1. In comparison with the full-fledged LOTH, we note that the presence of the balance factor $\alpha$ is beneficial for network performance, especially for early exit. Take MobileNetV1 as an example, the performance of Exit1 gains 0.99 with the balancing factor $\alpha$, which dramatically facilitates the learning ability of early exits with the low-capacity. In short, the mutual interplay between these components makes LOTH what it is.

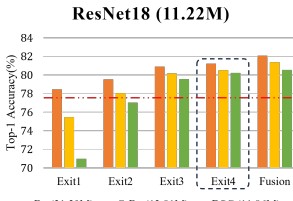 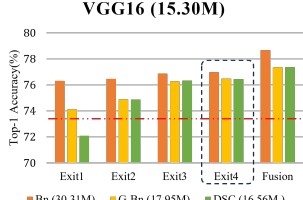 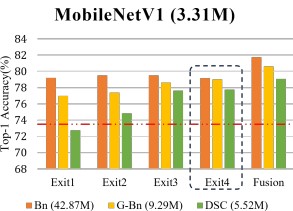 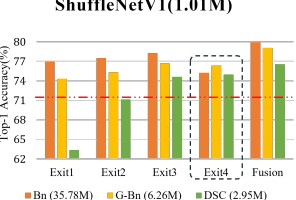

**Figure 4: The effect of different auxiliary blocks for model performance on the CIFAR-100, where the red dotted line indicates the baseline. The bracketed values indicate the total trained parameters of multi-exits and the backbone networks.**

## 5 CONCLUSION

In this paper, a novel self knowledge distillation framework via virtual teacher-students mutual learning called LOTH has been proposed, which focuses on fully exploiting the complementary knowledge of early exits to further enhance the effectiveness of distillation. Extensive experimental evaluations on CIFAR and Tiny-ImageNet have demonstrated that LOTH enables joint performance improvement of multi-exits. In addition, the impact of different sizes of auxiliary exits on model performance is sufficiently explored in this paper, and the two well-designed building blocks are verified to have the ability of balancing effectiveness and efficiency in self knowledge distillation. However, in this research, only logits are used to learn the knowledge of network, so the classification accuracy is still far from the requirements of applications. In the future, the incorporation of other types of knowledge like relations among objects will be researched to improve the ability of feature representation of network.

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
