# OpenReview forum: "Live On the Hump: Self Knowledge Distillation via Virtual Teacher-Students Mutual Learning"
_acmmm.org/ACMMM/2024/Conference — MM2024 Poster_

### Official Review · Reviewer_KYZr · 2024-05-23

**Rating:** 4
**Confidence:** 4

**Summary:**

The paper introduces a novel framework for self-knowledge distillation called LOTH, which addresses inefficiencies in traditional methods by effectively utilizing knowledge from shallow exits and auxiliary structures. LOTH employs a virtual teacher, constructed from the network's feature maps at each exit, to guide the learning process through mutual interaction between exits and the teacher. This process is facilitated by a well-designed loss function that enhances both the teacher's and the student's learning outcomes. Experiments demonstrate that LOTH outperforms traditional methods, making it suitable for resource-limited environments.

**Strengths:**

1. LOTH conducts extensive experiments and explorations across various datasets such as CIFAR-100 and Tiny-ImageNet, and on multiple network architectures, demonstrating its effectiveness and robustness. These experiments not only highlight LOTH's advantages in improving classification accuracy but also prove its potential for application on resource-constrained devices.
2. The proposed method is technically sound.

**Limitations:**

1. The paper could benefit from a more detailed exploration of the challenges associated with current sub-distillation methods. The employment of intermediate layers and additional trainable parameters for enhancing the student model has been extensively explored. It would be helpful if the specific advancements or contributions of this study were more clearly outlined to distinguish it from existing literature.
2. "use $\mathcal{E}_1$ to denote the auxiliary exit in the first stage" is described in terms of the network architecture's different depths, rather than stages of the training process. Therefore, this description might introduce some ambiguity.
3. The effectiveness of the approach has not been validated on larger datasets or more complex network architectures.

**Suitability:**

2

---

### Official Review · Reviewer_XpZq · 2024-05-24

**Rating:** 4
**Confidence:** 3

**Summary:**

The authors introduce LOTH (Live On the Hump), a self-knowledge distillation framework that uses virtual teacher-student mutual learning to enhance the performance of convolutional neural networks (CNNs). LOTH integrates knowledge from multiple network exits especially the early exits to create a virtual teacher, and applies a bidirectional mutual learning method between the virtual teacher and students. Moreover, they customize two well-designed auxiliary blocks to balance the efficiency and effectiveness based on the size of the network, leading to improved efficiency and effectiveness, as demonstrated through experiments on CIFAR-100 and Tiny-ImageNet datasets.

**Strengths:**

1.Proposes a novel self-knowledge distillation framework that leverages mutual learning between virtual teachers and network exits.

2.Effectively balances efficiency and effectiveness with adaptive auxiliary building blocks.

3.Provides a comprehensive evaluation against state-of-the-art distillation methods, showing superior performance.

**Limitations:**

1.This paper relies heavily on empirical results without deeply exploring theoretical underpinnings.

2.The integration of logits from multiple exits might introduce redundancy and computational overhead.

3.Some comparisons with existing methods lack detailed discussion on the potential reasons for performance differences.

4.The proposed method's applicability is only demonstrated on image classification task, many other tasks remain unexplored.

**Suitability:**

3

---

### Official Review · Reviewer_T9ji · 2024-05-26

**Rating:** 3
**Confidence:** 3

**Summary:**

The paper is about a novel self-knowledge distillation framework named "Live On the Hump" (LOTH), which is designed to address limitations in current self-knowledge distillation methods. These limitations include not fully utilizing the knowledge from shallow exits and neglecting the impact of auxiliary exits' structure on network performance. LOTH introduces a virtual teacher-student mutual learning approach to enhance the learning process.

The authors propose the LOTH framework that constructs a knowledgeable virtual teacher from the rich feature maps of each exit to aid in the learning of each exit. The framework incorporates the logit knowledge of each exit to guide the learning of the virtual teacher, promoting mutual learning through a well-designed loss function.

**Strengths:**

The design of two distinct auxiliary building blocks (Bottleneck and Group-Bottleneck) tailored for different network sizes is an effective way to balance efficiency and effectiveness. The theoretical foundation of mutual learning between a virtual teacher and multiple exits is sound and provides a robust framework for knowledge distillation. The method of feature fusion through global and local feature contexts is technically well-thought-out and aims to capture a comprehensive representation of the data.

**Limitations:**

Using shallow features as auxiliary labels in self-distillation isn't a novel concept. The experiments conducted on CIFAR-100 and Tiny-ImageNet are unconvincing, raising doubts about the model's scalability in terms of data and architecture. Additionally, it's unclear whether this model is applicable to transformers. Overall, the paper lacks richness in its content.

**Suitability:**

2

---

### Official Review · Reviewer_AX15 · 2024-05-27

**Rating:** 2
**Confidence:** 3

**Summary:**

This paper proposes a self knowledge distillation framework named LOTH is proposed, which views each exit as a student and fuses knowledge from auxiliary exits to build a knowledgeable virtual teacher. The bidirectional mutual learning between the virtual teacher and students contributes significantly to the capabilities of multi-exits learning。This paper designed an efficient adaptive feature fusion to adequately extracts important knowledge from each exit, which enhances the feature maps by channel and spatial attention,resulting in a powerful and knowledgeable virtual teacher.

**Strengths:**

1. The sentence structure of the paper is fluent and there are no obvious grammar or spelling errors
2. The paper conducted comparative experiments and ablation experiments, and the results are supported.
3.The paper provides a new approach to virtual teacher-student mutual learning through distillation networks, which is innovative.

**Limitations:**

1. The article lacks an explanation of the specific process of generating a virtual teacher in Figure 3. Please add more information.
2. Please provide a detailed explanation of whether the training and testing samples for CIFAR-100 in the 4.1.1 dataset were randomly selected or manually screened.
3. In the introduction, it is mentioned that LOTH can achieve higher accuracy with fewer resources, but this conclusion was not reflected in subsequent experiments.
4. The datasets used by the author are rather limited, especially lacking large-scale datasets such as comparisons on ImageNet, which results in a lack of reliability in the findings of this study.
5. Similarly, the author did not employ larger or newer models to validate the algorithm's performance. While this may be due to hardware limitations, it also somewhat constrains the credibility of this study.
6. The author needs to provide a more detailed description of the motivation and algorithm of this study to enhance its reliability.

**Suitability:**

2

---

### Meta-Review · Area_Chair_PRYg · 2024-07-07

**Recommendation:** Accept (Poster)
**Confidence:** 4

**Metareview:**

This paper proposes Live On the Hump (LOTH), a self knowledge distillation method with virtual teacher and mutual learning for knowledge distillation within a network. Specifically, multiple exits are utilized to generate a virtual teacher, along with other modules such as bidirectional mutual learning, adaptive feature fusion, and different scales of auxiliary exits. Experiments are conducted on CIFAR100 and Tiny-ImageNet to verify the effectiveness.

The initial ratings are Weak Reject, Borderline Reject, Borderline Accept, and Borderline Accept. The common concerns are the novelty, lack of larger dataset and larger network for evaluation. In addition, T9ji challenged the suitability for MM submission.

The authors provide a solid rebuttal to address the concerns and suitability issue. After rebuttal, two reviewers acknowledge the authors efforts to address the concerns of lacking larger networks/datasets, and the suitability is also modified. Therefore, the two reviewers raised their ratings to Borderline Accept. Other two reviewers remain their rating. So finally, all scores are Borderline Accept.

After reading the paper, rebuttal, and all comments, I agree with the reviewers that this paper should be accepted.
Therefore, I recommend to Accept this paper as Poster.